# The Interplay between Anti-Angiogenics and Immunotherapy in Colorectal Cancer

**DOI:** 10.3390/life12101552

**Published:** 2022-10-06

**Authors:** Brigida Anna Maiorano, Alessandro Parisi, Evaristo Maiello, Davide Ciardiello

**Affiliations:** 1Oncology Unit, IRCCS Foundation Casa Sollievo della Sofferenza, 71013 San Giovanni Rotondo, Italy; 2Department of Translational Medicine and Surgery, Catholic University of the Sacred Heart, 00168 Rome, Italy; 3Department of Oncology, Università Politecnica delle Marche, Azienda Ospedaliero-Universitaria, Ospedali Riuniti di Ancona, 60126 Ancona, Italy; 4Department of Life, Health and Environmental Sciences, University of L’Aquila, 67100 L’Aquila, Italy; 5Medical Oncology Unit, Department of Precision Medicine, “Luigi Vanvitelli” University of Campania, 80131 Naples, Italy

**Keywords:** colorectal cancer, CRC, anti-angiogenics, angiogenesis, immune checkpoint inhibitors, ICIs, immunotherapy, bevacizumab, pembrolizumab, nivolumab

## Abstract

**Simple Summary:**

Colorectal cancer is a frequent and lethal neoplasm. The tumor often creates new vessels to grow and spread—a process called ‘angiogenesis’. Therefore, drugs blocking angiogenesis are effective against this malignancy. On the other side, immune checkpoint inhibitors, which unleash the immune system to fight against tumors, have limited efficacy in patients carrying instability of DNA regions called microsatellites. However, there is an interaction between angiogenic factors and the immune system. This gives a chance to combine anti-angiogenic agents and immune checkpoint inhibitors to improve the efficacy of treating this malignancy.

**Abstract:**

Angiogenesis, a hallmark of cancer, plays a fundamental role in colorectal cancer (CRC). Anti-angiogenic drugs and chemotherapy represent a standard of care for treating metastatic disease. Immune checkpoint inhibitors (ICIs) have changed the therapeutic algorithm of many solid tumors. However, the efficacy of ICIs is limited to mCRC patients carrying microsatellite instability (MSI-H), which represent approximately 3–5% of mCRC. Emerging evidence suggests that anti-angiogenic drugs could exhibit immunomodulatory properties. Thus, there is a strong rationale for combining anti-angiogenics and ICIs to improve efficacy in the metastatic setting. Our review summarizes the pre-clinical and clinical evidence regarding the combination of anti-angiogenics and ICIs in mCRC to deepen the possible application in daily clinical practice.

## 1. Introduction

Colorectal cancer (CRC) represents the third most common cancer and the second leading cause of cancer-related deaths worldwide, with an increasing incidence in the last decades [1]. Around 20% of patients are diagnosed in the metastatic stage; moreover, 50% of patients with localized disease will develop metastases eventually [2].

Starting from the late 1950s, when 5-fluorouracil (5-FU) was tested, the therapeutic scenario of metastatic CRC (mCRC) evolved from monotherapy to combinations of 5-fluorouracile (5-FU), oxaliplatin, and irinotecan as doublets or triplets; this significantly moved forward mCRC survival. A subsequent fundamental improvement for mCRC patients derived from a personalized approach, consisting of administering agents targeting specific mutations, such as epidermal growth factor receptor (EGFR), B-Raf, and v-Raf murine sarcoma viral oncogene homolog B (BRAF), or anti-angiogenic drugs [2,3]. Effectively, angiogenesis has been indicated as one of the hallmarks of cancer, involved in the development and spread of CRC [4]. Starting in the 2000s, the clinical development of anti-angiogenic drugs, including bevacizumab, ramucirumab, aflibercept, and regorafenib, represented a benchmark for the treatment of mCRC [5,6,7,8,9,10,11,12,13,14,15]. Presently, anti-angiogenics can be used in both the first and subsequent lines of mCRC treatments. Bevacizumab is a monoclonal antibody directed against vascular endothelial growth factor A (VEGF-A). In several phase III trials, when bevacizumab was added to first-line chemotherapy, progression-free survival (PFS) was prolonged up to 12 months, and overall survival (OS) reached 31 months [6,7,8,9]. Bevacizumab was also effective in later lines after the progression to front-line treatment reaching around one year of OS [13]. The other anti-angiogenics have been tested after progression to first-line therapy in patients that also received bevacizumab. Aflibercept is a fusion protein directed against VEGF, and ramucirumab binds VEGF-Receptor 2 (VEGFR-2); they determined an OS of around 13 months combined with FOLFIRI in pre-treated patients [14,15]. Furthermore, the tyrosine kinase inhibitor (TKI) regorafenib proved anti-tumor activity in patients with refractory mCRC and is used as a standard of care (SOC) as a later-line treatment [16].

Immunotherapy profoundly transformed the therapeutic scenario of several malignancies; in fact, in the last two decades, the use of immune checkpoint inhibitors (ICIs) spread in many solid tumors after achieving meaningful improvements in survival and quality of life [17,18,19,20,21,22,23,24]. Finding predictive biomarkers for ICIs to allow a better patient selection remains one of the most critical unanswered questions in contemporary immune oncology. Microsatellite instability (MSI) has been addressed as a possible predictive biomarker for ICIs in mCRC. Microsatellites are repeated sequences of one to six nucleotides, often altered during DNA replication. The mismatch repair system (MMR) is a group of proteins (including MutL Homolog [MLH] 1, MLH3, MutS Homolog [MSH] 2, MSH6, MSH3, Protein homolog [PMS] 1, PMS2, and Exonuclease 1 [Exo1]) that can detect and repair microsatellites errors, thus maintaining genome integrity. Hence, an alteration of this system leads to MSI [25,26]. The accumulation of mutations eventually determines a high neo-antigen load, leading to the activation of the immune system, representing a possible explanation for the higher efficacy of ICIs in these patients compared to MS stable (MSS) [27]. Effectively, MSI-H tumors express higher levels of tumor-infiltrating lymphocytes (TILs), cytotoxic T-lymphocyte-associated protein 4 (CTLA4), programmed cell death 1 (PD1)/PD-ligand 1 (PD-L1), indoleamine 2,3-dioxygenase (IDO), and lymphocyte-activation gene 3 (LAG-3) [28].

Initial studies of ICIs in mCRC included pre-treated patients with MSI. Both KEYNOTE-016 and KEYNOTE-164 demonstrated the efficacy of pembrolizumab as monotherapy in chemo-resistant patients, with an overall response rate (ORR) up to 40% and over 31 months of mOS [29,30]. In phase III KEYNOTE-177 trial, patients were randomized to pembrolizumab or chemotherapy as first-line treatment. Pembrolizumab boosted ORR to 43.8% and doubled mPFS to 16.5 (vs. 8.2) months, and was therefore approved by the Food and Drug Administration (FDA) and European Medical Agency (EMA) [31]. Similar results were observed with nivolumab in pre-treated MSI patients [32]. Interestingly, a combination of Nivolumab plus the anti-CTLA4 Ipilimumab led to an increased response rate and durable response both in untread and refractory MSI mCRC [32,33].

However, MSI is observed only in 3–5% of mCRC. The vast majority of patients with proficient MMR or microsatellite stable (MSS) tumors do not benefit from ICIs [34]. In this scenario, different combinatory strategies have been investigated to convert MSS ‘cold tumors’ into ‘immune-competent malignancies’, and therefore are amenable to benefit from immunotherapy. Notably, anti-angiogenic drugs are associated with immunomodulatory properties [35,36]. As a consequence, an exciting possibility is represented by the association of anti-angiogenic with immunotherapeutic agents. Our review aims to summarize the available evidence of this combination to define better the pre-clinical rationale that underlies this novel treatment strategy to elicit an immune response in MSS mCRC patients, its clinical application in daily clinical practice, and future directions for research in this field.

## 2. Rationale to Combine Anti-Angiogenics and ICIs in mCRC

Angiogenesis is a complex process of forming new blood vessels that differentiate from existing endothelial cells (ECs), used as a mechanism of growth and spread by all types of solid tumors [37]. Six ligands and three receptors constitute the VEGF pathway involved in the angiogenetic process regulation [38]. Hypoxia, which often characterizes solid tumors, leads to the activation of the hypoxia-inducible factors (HIF), leading to the transcription of genes, VEGF-A included, aiming to ensure adequate tissue oxygenation [39]. The tumor cells themselves can produce VEGF-A, which binds VEGFR-2 of nearby blood vessels and stimulates the differentiation and growth of EC: the ECs’ shift from a dormant to an active state has been described as the ‘angiogenic swift’ [5]. These changes occur in blood vessels and extracellular matrix, with vasodilatation, permeabilization, and EC migration leading to the formation of new blood vessels [37]. The increased production of VEGF-A triggered by the hypoxic stimulus during neo-angiogenesis shifts the tumor microenvironment (TME) towards immune suppression. VEGF-A can inhibit the maturation and function of dendritic cells (DCs), upregulating PD-L1 expression on DCs, and finally induce T-cell suppression [40,41,42,43]. The abnormal architecture of new blood vessels induces an increase in interstitial fluid pressure that, together with the lack of adhesion molecules (such as vasculature cell adhesion molecule [VCAM]-1), reduces the TILs infiltration at tumor sites. Moreover, hypoxia regulates some immune suppressive signals, such as PD-L1, IDO, interleukin-6 (IL-6), and IL-10. It induces the upregulation of chemokines such as chemokine (C-C motif)-ligand (CCL)-22 and CCL28, that recruit Tregs at tumor sites, altering the equilibrium between T-effectors and Tregs [43,44,45]. Another effect of hypoxia is macrophage polarization towards the M2-like phenotype. Furthermore, ECs express Fas ligand (FasL) that reduces CD8^+^ T cells but not Tregs, as the latter express FLICE-inhibitory protein (c-FLIP) [43]. As a result, the TME balance is shifted towards immune suppression. The combined restoration of immune responsiveness induced by ICIs and inhibition of angiogenesis after anti-angiogenic therapy could revert this effect on the TME and restore the reciprocal efficacy of the two treatments (Figure 1).

The increased production of vascular endothelial growth factor A (VEGF-A), triggered by the hypoxic stimulus during neo-angiogenesis, shifts the tumor microenvironment (TME) towards immune suppression. VEGF-A inhibits the maturation and function of dendritic cells (DCs), upregulating programmed death ligand 1 (PD-L1) expression on DCs, and finally induces T-cell suppression. Another effect of hypoxia is macrophage polarization towards the M2-like phenotype. Moreover, hypoxia regulates some immune suppressive signals, such as PD-L1, indoleamine 2,3-dioxygenase (IDO), interleukin-6 (IL-6), IL-10, and induces the upregulation of chemokines such as chemokine (C-C motif)-ligand (CCL)-22 and CCL28, that recruit Tregs at tumor sites, altering the equilibrium between T-effectors and Tregs. The abnormal architecture of new blood vessels induces an increase in interstitial fluid pressure that, together with the lack of adhesion molecules (such as vasculature cell adhesion molecule [VCAM]-1), reduces the TILs infiltration at tumor sites. Moreover, endothelial cells express Fas ligand (FasL) that reduces CD8+ T cells but not Tregs, as the latter express FLICE-inhibitory protein (c-FLIP). As a result, the TME balance is shifted towards immune suppression.

## 3. Clinical Trials of Anti-Angiogenics Plus ICIs in mCRC

As the potential synergism between anti-angiogenics and ICIs was effective and was approved in several solid tumors, several trials were also carried out in mCRC. As ICIs efficacy was restricted to MSI-H patients, many combination trials focused on MSS subjects. Most studies included pre-treated patients, but only five trials were conducted on naïve patients. (Table 1).

In the phase Ib NCT01633970 study, atezolizumab was co-administered with bevacizumab in pre-treated mCRC patients (Arm A), and with bevacizumab and FOLFOX in naïve (Arm B) mCRC patients. Patients were not selected for RAS-mutational status. ORR was 8% among the 13 patients of Arm A, and 36% among the 26 patients in Arm B. G3 or more AEs occurred in 64% of Arm A and 73% of Arm B patients [46]. In the MODUL trial, 445 BRAF wild-type patients received 5-FU + bevacizumab with or without atezolizumab as maintenance after induction with first-line FOLFOX + bevacizumab. Although ORR and DCR were numerically but not statistically significantly higher in the experimental arm compared with SOC, no difference in PFS or OS was observed [47]. Biomarker analyses are currently ongoing and will be presented. In the phase II BACCI trial (NCT02873195), 133 heavily pre-treated MSS mCRC patients were randomized to capecitabine plus bevacizumab plus/minus atezolizumab. Patients could have progressed on previous chemotherapy and anti-VEGF agents in case of RAS mutations. In the atezolizumab group, mPFS was 4.4 months vs. 3.6 in the PBO group. ≥G3 AEs occurred as hypertension (7 vs. 4.3%), diarrhea (7 vs. 4.3%), and hand-foot syndrome (HFS) (7 vs. 4.3%). An exploratory analysis showed that patients without liver metastases had a higher ORR than those with liver metastases (23.1% vs. 5.8%) and tended to have better PFS and OS [48].

In the AtezoTRIBE trial, a multicenter phase II study, naïve mCRC patients were randomized 1:2 to FOLFOXIRI plus bevacizumab without or with atezolizumab, independently from RAS or BRAF mutational status. PFS was the primary endpoint. At the data cut-off, 73 patients had received the standard treatment and 145 atezolizumab. The trial met its primary endpoint, as atezolizumab improved mPFS to 13.1 vs. 11.5 months (HR 0.69, 95% CI 0.56–0.85; *p* = 0.012). OS data are still immature. The most frequent ≥G3 AEs were neutropenia and diarrhea in both groups. The authors conducted a translational analysis to identify possible biomarkers of response. Interestingly, patients with higher tumor mutational burden (TMB) and Immunoscore IC showed a prolonged PFS [49].

CheckMate 9X8 is a randomized phase II study that investigates the addition of nivolumab to the standard of care (SOC—FOLFOX + bevacizumab) as a first-line treatment for mCRC, independently from RAS, BRAF, or MS status [50]. The primary endpoint was not met, as mPFS was 11.9 months in both experimental and SOC arms. However, adding nivolumab to FOLFOX + bevacizumab correlated with a higher ORR (60% vs. 46%) and more durable responses than SOC. These data suggest that there is a subset of patients that could benefit from ICIs.

The NIVACOR trial is a single-arm, phase II study evaluating the combination of nivolumab with FOLFOXIRI + bevacizumab followed by maintenance with bevacizumab plus nivolumab in patients with RAS/BRAF mutant untreated mCRC [51]. Of the 73 patients enrolled, 10 were MSI, and for 11 patients, microsatellite instability was not assessed. ORR was 76.7% in the overall population, with a DCR of 97.3%; 2 (2.7%) pts were not evaluable. The mDOR was 8.4 (95%CI, 7-NE) months. In the subset of MSS patients, ORR was 78.9% with an mDOR of 7.59 (95% CI 6.21–11.43) months, DCR of 96.2%, and mPFS of 9.8 (95%CI 8.18–15.24) months.

In a single-center phase II trial, 44 MSS mCRC patients with SD or PD on previous fluoropyrimidine-based therapy received capecitabine plus bevacizumab plus pembrolizumab. Patients were enrolled independently from RAS/BRAF status. ORR was 5%, mPFS 4.3 months, 6 mos PFS 31.1%, mOS 9.6 months. ≥G3 AEs occurred in 28% of patients, with the requirement of dose reduction/interruption in 58% of cases [52].

In the NCT03050814 phase II trial, patients with untreated MSS mCRC (independently from RAS/BRAF status) were randomized to mFOLFOX6 + bevacizumab with or without avelumab plus a CEA-targeted vaccine. No differences emerged between the two arms regarding PFS (HR = 1.06, 95% CI, 0.38–2.96; *p* = 0.91), and ORR was 50% in both groups [53].

There is emerging evidence that regorafenib with ICIs could exert an anti-tumor activity by various mechanisms, including the activation of the immune system [54]. In this regard, OU and colleagues observed that regorafenib could influence the polarization of tumor-associated macrophages. Based on this rationale, different studies evaluated the combination of regorafenib with ICIs.

Nivolumab was tested with regorafenib in two phase Ib and one phase II studies. In the phase Ib REGONIVO (EPOC1603) trial, 25 pre-treated mCRC patients were included, independently from RAS/BRAF mutational status. ORR was 36%, and mPFS was 7.9 months. Patients with lung metastases tended to have better outcomes compared with liver metastases [55]. Among 51 patients with mismatch repair proficient (pMMR) mCRC of NCT03712943 study, 10% achieved PR, 53% SD with an mPFS of 4.3 months, and an mOS of 11.1 months. The most common G3/G4 AEs were hypertension (16%), rash (19%), and anemia (6%) [56]. Among 70 mCRC patients of the NCT03712943 study, an ORR of 21.7% was achieved. Higher baseline levels of cytotoxic T cells, FoxP3+ Tregs, and macrophages tended to better outcomes. Lower plasma levels of vascular biomarkers such as VEGF-D, Angiopoietin-2 (Ang-2), and von Willebrand factor (VWF) correlated with longer PFS. The most common G3/G4 AEs were rash (14%), fatigue (7%), pneumonia (6%), and increased bilirubin (6%) [57]. In the phase I/II NCT03657641 study, 73 patients with MSS mCRC were treated with pembrolizumab plus regorafenib, reaching an mPFS of 4.3 months and an ORR of 0%, with 49% of patients having SD. ≥G3 rash occurred in 20% of patients, ≥G3 HFS in 7% [58].

In the REGOMUNE (NCT03475953) phase II trial, 48 patients received the combination of regorafenib and avelumab 10 mg/kg q2w. Patients were eligible if MSS, but independently from RAS/BRAF mutational status. mPFS was 3.6 mos, mOS 10.8 mos. PPES (29.8%), hypertension (23.4%), and diarrhea (12.8%) were the most common >G3 AEs. High infiltration of tumor-associated macrophages (TAMs) at baseline was significantly associated with shorter PFS (1.9 vs. 3.7 mos; *p* = 0.045) and OS (4.8 mos vs. NR, *p* = 0.027). On the contrary, increased CD8+ after treatment starting was significantly associated with better PFS (*p* = 0.011) [59].

In a phase Ib/II study, the safety and activity of regorafenib plus the anti-PD1 toripalimab were tested in 42 MSS pre-treated mCRC patients, independently from RAS/BRAF status [60]. The ORR was 15.2% and the DCR was 36.4% in evaluable patients with recommended phase II dose (80 mg regorafenib plus toripalimab). mPFS and mOS were 2.1 months and 15.5 months, respectively. Similarly to previous findings, patients with liver metastases exhibited lower ORR than those without (8.7% versus 30.0%).

In the LEAP-005 phase II study (NCT03797326), patients with MSS/pMMR (but independently from RAS/BRAF status) mCRC were treated with pembrolizumab 200 mg q3w plus lenvatinib 20 mg daily. Among the 32 treated patients, ORR was 22%, and 50% of patients experienced AEs, with three treatment discontinuation. DCR was 47%, mDOR was not reached (NR), mPFS 2.3 months, and mOS 7.5 months [61]. Based on these results, an ongoing randomized phase III study evaluating pembrolizumab plus lenvatinib in pre-treated patients with mCRC is ongoing [62].

In the NCT03239145 phase Ib trial, 18 MSS (independently from RAS/BRAF status) heavily pre-treated mCRC patients received pembrolizumab plus trebananib, an anti-Ang 1/2 antibody. Ang-2 is produced by ECs. DCR was 33%, median time-to-progression (TTP) was 2.8 months, and mOS 9 months. The most common AEs of the combination were diarrhea, limber edema, proteinuria, and transaminase increase; only two pembrolizumab-related G3/G4 AEs were reported (pneumonitis and transaminase increase) [63,64].

## 4. Discussion

Angiogenesis represents a hallmark of cancer, also in mCRC; therefore, anti-angiogenic drugs are regularly used in clinical practice in different settings [4,6,7,8,9,10,11,12,13,14,15,16,65]. On the other hand, even if immunotherapy is one of the significant achievements of modern oncology, in mCRC its use is still limited to MSI-H patients. Based on the potential interaction with the immune response, there is a rationale to combine anti-angiogenic with ICIs. Despite the solid biological rationale and robust pre-clinical evidence, further studies are needed to find the best combinatory strategies and potential biomarkers of response to improve patients’ selection.

Endothelial cells (ECs) share common ancestors with immune cells, which is why they play a role in immune modulation, acting as a sort of gatekeeper controlling the passage of patrolling immune cells from circulation into tissues [66,67,68]. E-selectin and P-selectin on ECs interact with T-cell ligands. T cells attracting chemokines, such as CCL2 and CXCL9-10-11, interact with their receptors on T-cells (such as CCL2 receptor [CCR2] and CXC receptor 3 [CXCR3]), activating them [67,68,69]. After activation, T-cell integrins interact with surface adhesion molecules, very late antigen-4 (VLA4) with vascular cell adhesion molecule 1 (VCAM1), lymphocyte function-associated antigen 1 (LFA-1) with intercellular adhesion molecule (ICAM) [67,68,70]. ECs activation and expression of adhesion molecules can be induced by cytokines, such as IL-6, IL-1b, IFNγ, and TNFα, or pathogen-associated molecular patterns, such as lipopolysaccharide. Inflammatory cytokines, and the same VEGF, act as proangiogenic but also immunomodulatory molecules. For example, VEGF inhibits the expression of surface adhesion molecules and T cells recruiting chemokines such as CXCL9-10-11; moreover, it induces the ECs expression of FasL that causes T-cell apoptosis and recruits Tregs. Therefore, during inflammation, ECs can recruit different immune cells, such as T-effectors, monocytes, and neutrophils [66,67,68,71]. Furthermore, as tumor vessels are structurally and functionally abnormal, they contribute to immune suppression by implementing necrosis-hypoxia-acidosis. In fact, the production of immune suppressive lactate, nitric oxide, and reactive oxygen species suppresses T-effectors. Moreover, MDSCs and Tregs are recruited, and macrophages shift towards an M2-like subtype, reducing the activity of cytotoxic T cells. Furthermore, ECs can express inhibitory checkpoints such as PD-L1/2, IDO, and T-cell immunoglobulin and mucin domain-containing protein 3 (TIM3), even leading to T-cell death or anergy [66,67,68]. Finally, a shift towards major histocompatibility complex (MHC)-I overexpression and MHC-II decrease can be associated with the lack of co-stimulatory molecules, such as CD80/CD86, and a higher immune tolerogenicity [66,67,68,72]. As a result, the inhibition of tumor angiogenesis through anti-angiogenic drugs could contribute to a more immune-responsive TME and act in synergy with ICIs. It has been previously demonstrated that there is an interplay between T cells and tumor vascularization inducing CD4+ T-cell activation, IFNγ production, and subsequent boosting angiogenesis homeostasis, but also immune response [66,67,68] (Table 2).

To date, the combination of chemotherapy plus ICIs appears less effective than expected within most studies, with no clear advantage over standard treatment. Nevertheless, in the AtezoTRIBE study, adding PD-1/PD-1 blockade to the intensive chemotherapy FOLFOXIRI plus bevacizumab seemed to prolong PFS compared with SOC [49]. The authors conducted an exploratory analysis to investigate the role of different potential biomarkers, including the microsatellite instability status, tumor mutational burden (TMB), and Immunoscore IC, as predictive of ICIs response. Interestingly, among patients with MSS tumors, those with Immunoscore IC and TMB high displayed a prolonged PFS. With the limits of a small number of patients included in the biomarker analysis, these results could be considered hypothesis-generating for future investigation. Another study conducted on 18 CRC patients, and CRC cell lines, showed a higher expression of CTLA4 in CRC tissues compared to adjacent non-CRC ones and that this expression could be altered after administering capecitabine, opening up the way for further investigations regarding treatment combinations for improving ICIs efficacy in this malignancy [73]. Effectively, angiogenesis itself can interact with TME, inducing a shift toward immune suppression: DCs are reduced and upregulated the expression of tolerogenic signals such as PD-L1, T cells are suppressed, and TILs infiltration is reduced at tumors sites [40,41,42,43]. Immune suppression is further potentiated by hypoxia, which reinforces immune suppressive signals, PD-L1, IDO, IL-6, IL-10, recruits Tregs, and stimulates macrophage polarization towards M2-like rather than M1-like subtype [43,44,45].

Effectively, an immune suppressive TME has a crucial role in CRC liver metastasis (CRC-LM) [60,74,75]. A weakened liver’s immune-killing ability promotes the development of CRC-LM. Thus, the overexpression of immune checkpoints (PD1/PD-L1), immunosuppressive cytokines such as transforming growth factor (TGF)-beta and IL-10, and the subsequent activation of Tregs and TAMs, lead to an immunosuppressive TME and favor the CRC-LM growth. In a monocenter retrospective study, the prognostic role of CRC-LM was assessed in a cohort of 95 patients with MSS refractory mCRC treated with ICIs [74]. The ORR in the overall population was 8.4% (8/95). Interestingly, 8 out of 41 (19.5%) patients without CRC-LM achieved a CR/PR, whereas no response was observed in 54 patients with liver metastases. The DCR was 58.5% (24 out of 41) in patients without liver metastases and 1.9% (1 out of 54) in patients with liver metastases. Moreover, patients without CRC-LM displayed a statistically significant increase in PFS compared with patients with CRC-LM (4.0 vs. 1.5 months; *p* < 0.001). On the same line, different studies investigating the use of TKIs with ICIs showed that patients with CRC-LM were less likely to respond to combinatory strategies [55,59]. Considering the small number of trial patients, this observation should be taken with caution. The ongoing randomized phase III study LEAP-17 will lighten the efficacy of pembrolizumab plus lenvatinib in this subset of patients [61]. Similarly, further attempts to combine anti-angiogenic/TKI with ICIs and chemotherapeutic regimens are ongoing at different stages. Their results could better clarify the efficacy of the combined mechanisms and eventually improve the therapeutic options in daily clinical practice (Table 3). Up to now, no particular safety concerns have emerged from the combination studies, as the primary toxicity derives from the administered anti-angiogenics. In contrast, ICIs do not seem to raise specific safety concerns. Ongoing studies will also shed light on the safety profile of the different agents when used in combination. Finally, beyond the RAS/BRAF mutational status, which does not seem to influence the response to this combination, and MSS status, a deep investigation regarding biomarkers for treatment response should be conducted in order to allow an optimal patients selection towards a tailored therapeutic approach, combined with sequencing strategies.

An emerging amount of evidence indicates that the gut microbiome could regulate the homeostasis of different physiological conditions. Alteration of the composition and biodiversity of gut microbiota, a condition called dysbiosis, is involved in different pathological conditions, including CRC [76]. Although the role of specific microbes in modulating the efficacy and tolerability of immunotherapy has been addressed in different malignancies, little evidence is currently available for mCRC patients [77,78,79]. To assess the potential role of gut microbiota in response to ICIs, we conducted a retrospective analysis on available pre-treatment stool samples of mCRC and non-small cell lung cancer (NSCLC) patients treated with cetuximab plus avelumab [80,81,82]. Fascinatingly, in five long-term responding patients with MSS mCRC, PFS (9–24 months) was significantly increased in two butyrate-producing bacteria, *Agathobacter* M104/1 (*p* = 0.018) and *Blautia* SR1/5 (*p* = 0.023) compared with nine patients with shorter PFS (2–6 months). These results were consistent with the validation cohort of NSCLC patients that received the combination of cetuximab and avelumab. In the phase Ib/II study evaluating the combination of regorafenib plus toripalimab, a gut microbiome analysis of the baseline fecal samples was performed [83]. Notably, a significantly increased relative abundance and positive detection rate of *Fusobacterium* was observed in non-responders compared with responders. Moreover, patients with a high abundance of *Fusobacterium* exhibited a shorter PFS than those with low abundance (mPFS = 2.0 vs. 5.2 months; *p* = 0.002).

## 5. Conclusions

Following the robust results of blocking the angiogenesis and PD1/PD-L1 axis in hepatocarcinoma and renal cell carcinoma, there was great interest in this therapeutic strategy for CRC. Unfortunately, preliminary results were more disappointing than expected. Whereas a small subset of mCRC is experiencing tumor regression with ICIs plus anti-angiogenic drugs, most patients do not benefit from the treatment. The commonly used molecular classifications are not prognostic for this combination of treatments. Therefore, further translational studies are needed to identify clinical, immunological, and molecular predictive biomarkers of response.

## Figures and Tables

**Figure 1 life-12-01552-f001:**
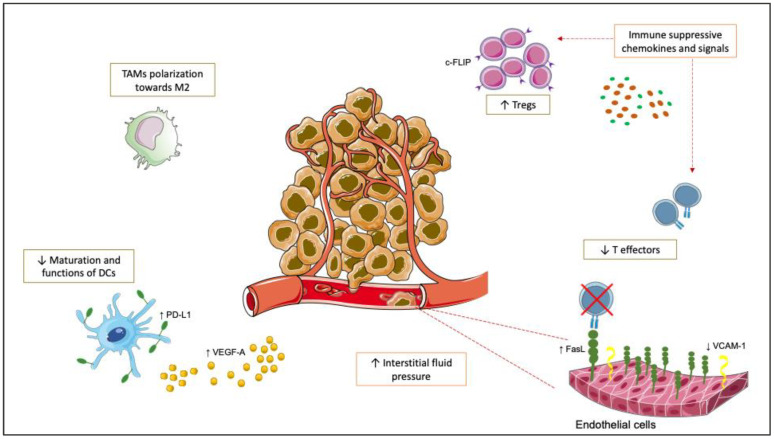
Effects of hypoxia/neo-angiogenesis on tumor microenvironment.

**Table 1 life-12-01552-t001:** Clinical trials of anti-angiogenics plus ICIs in mCRC.

Trial Name	First Author	Year	Phase	Nr. of Patients	Treatment	ORR, *%*	mPFS, Months	mOS, Months	Safety
NCT01633970	Bendell	2015	1b	Arm A (pre-treated): 13 Arm B (naïve): 26	Arm A: atezolizumab + bevacizumab Arm B: atezolizumab + FOLFOX + bevacizumab	Arm A: 8% Arm B: 36%	NA	NA	≥G3 AEs: 64% (Arm A), 73% (Arm B)
MODUL (NCT02291289)	Grothey	2018	2	445 (naïve, BRAF wt)	Maintenance bevacizumab +/− atezolizumab after FOLFOX + bevacizumab	NA	Not met	Immature data	NA
CheckMate 9X8 (NCT03414983)	Lenz	2022	2	Experimental Arm: 127 Control Arm:68	Experimental Arm: FOLFOX + Bevacizumab + Nivolumab Control Arm: FOLFOX + Bevacizumab	Experimental Arm: 60% vs Control Arm:48%	11.9 months in both Arm	Immature data	Grade 3−4 AEs 75% experimental Arm Vs. 48% control Arm
BACCI (NCT0287319)	Mettu	2019	2	133 (pre-treated)	Capecitabine and bevacizumab + atezolizumab or placebo	NA	4.4 vs. 3.6	NA	≥G3 AEs: hypertension (7 vs. 4.3%), diarrhea (7 vs. 4.3%), HFS (7 vs. 4.3%).
Atezo TRIBE (NCT03721653)	Antoniotti	2022	2	218 (naïve)	FOLFOXIRI + bevacizumab +/− atezolizumab	NA	13.1 vs. 11.5 (*p* = 0.012)	NA	≥G3 AEs: neutropenia, diarrhea; 2 treatment-related deaths
CheckMate 9X8 (NCT03414983)		2022	2	195 (naïve)	FOLFOX + bevacizumab +/− nivolumab	60% vs. 46%	11.9 vs. 11.9	NA	≥G3 AEs 75% vs. 48%
NIVACOR (NCT04072198)	Damato	2022	2	73 (naïve, RAS/BRAF mut)	FOLFOXIRI + bevacizumab + nivolumab	76.7%	10.1 months	NA	≥G3 AEs: neutropenia, diarrhea, fatigue and hypertension
NCT03946917	Wang	2021	1b/2	42 (MSS pre-treated)	Toripalimab + regorafenib	15.2	2.1	15.5	≥G3 AEs: Hand-foot syndrome; Rash; impaired liver function
LEAP-005 (NCT03797326)	Gomez-Roca	2021	2	32 (MSS, pre-treated)	Pembrolizumab + lenvatinib	22	2.3	7.5	50% AEs
NCT03396926	Bocobo	2022	2	44 (MSS, pre-treated)	Pembrolizumab + capecitabine + bevacizumab	5	4.3	9.6	28% ≥G3 AEs, 58% dose reduction/interruption
NCT03050814	Redman	2022	2	26 (MSS, naïve)	mFOLFOX6 + bevacizumab +/− avelumab + CEA-targeted vaccine	50% vs. 50%	No differences	NA	NA
REGONIVO (EPOC1603)	Fukuoka	2020	1b	25 (pre-treated)	Nivolumab + regorafenib	36	7.9	NA	≥G3 AEs: rash (12%), proteinuria (12%), PPED (10%)
NCT03712943	Kim	2022	1b	51 (MSS, pre-treated)	Nivolumab + regorafenib	10	4.3	11.1	≥G3 AEs: hypertension (16%), rash (19%), anemia (6%)
NCT04126733	Fakih	2021	2	70 (MSS, pre-treated)	Nivolumab + regorafenib	21.7	15 weeks	52 weeks	≥G3 AEs: rash (14%), fatigue (7%), pneumonia (6%), increased bilirubin (6%)
NCT03657641	Barzi	2022	1/2	73 (MSS, pre-treated)	Pembrolizumab + regorafenib	0	2.8	9.6	≥G3 rash 20%, ≥G3 HFS 7%
REGOMUNE (NCT03475953)	Cousin	2021	2	48 (MSS, pre-treated)	Avelumab + regorafenib	0	3.6	10.8	≥G3 AEs: PPED (29.8%), hypertension (23.4%), diarrhea (12.8%)
(Wang et al.)		2021	1b/2	42 (MSS, pre-treated)	Toripalimab + regorafenib	15.2	2.1	15.5	≥G3 AEs: 38.5%
NCT03239145	Rahma	2020	1b	18 (MSS, pre-treated)	Pembrolizumab + trebananib	NA	NA	9	AEs: diarrhea, limber edema, proteinuria, transaminase increase

AEs: adverse events; BRAF: v-raf murine sarcoma viral oncogene homolog B1; FOLFOX: folinic acid, 5-fluorouracil, and oxaliplatin; FOLFOXIRI: folinic acid, 5-fluorouracil, oxaliplatin, irinotecan; HFS: hand-foot syndrome; ICIs: immune checkpoint inhibitors; mCRC: metastatic colorectal cancer; mPFS: median progression-free survival; mOS: median overall survival; MSI-H: microsatellite instability-high; MSS: microsatellite stability; NA: not available; NR: not reached; ORR: overall response rate; PPED: palmar-plantar erythrodysesthesia; wt: wild-type.

**Table 2 life-12-01552-t002:** Endothelial cells (ECs) as checkpoint for immunological patrolling. Receptors expressed by ECs, or circulating factors interacting with ECs, and relative functions in immunological patrolling are listed.

Molecule	Role
Chemokines (e.g., CCL2, CXCL4/10)	Attracting and binding immune cells with ECs
Circulating pro-inflammatory cytokines (IFNγ, TNFα)	Favoring the activation of ECs with exposure of cell adhesion molecules, immune modulation recruiting MDSCs, Tregs, macrophages shifting towards M2-like subtype
VEGF	Recruiting immune suppressive cells such as Tregs, inhibiting expression of cell surface adhesion molecules, reducing T cells recruiting chemokines, inducing FasL expression on ECs
Adhesion molecules (E-/P-selectin, VCAM, ICAM)	Recruiting and binding immune cells
MHC-I	Overexpression associated with lack of co-stimulatory molecules (CD80/CD86)
MHC-II	Decrease on tumor vessels, contributing to immune tolerogenicity
PD-L1/2	Creating an immune suppressive tumor microenvironment through the crosstalk between immune cells, cancer cells, and vessels
NO, ROS	Altering immune cells infiltration and suppressing CD8+ T cells
IDO, TIM3	After stimulation of ECs by cytokines such as IFNγ inducing T-cell death, cell cycle arrest, and anergy
FasL	Causing T-cell apoptosis

CCL2: chemokine (C-C motif) ligand 2; CD: cluster of differentiation; CXCL4/10: CXC chemokine ligand 4/10; ECs: endothelial cells; ICAM: intercellular adhesion molecule; IDO: Indoleamine 2,3-dioxygenase; IFNγ: Interferon gamma; MDSCs: myeloid-derived suppressive cells; MHC: major histocompatibility complex; NO: nitric oxide; PD-L1/2: programmed death ligand 1/2; ROS: reactive oxygen species; TIM3: T-cell immunoglobulin and mucin domain-containing protein 3; TNFα: tumor-necrosis factor alpha; VEGF: vascular endothelial growth factor; VCAM: vascular cell adhesion molecule.

**Table 3 life-12-01552-t003:** Ongoing trials of immune checkpoint inhibitors and anti-angiogenics combination.

Trial Identification	Phase	Drug Combination	Primary Endpoint
NCT03657641	I/II	Pembrolizumab + Regorafenib	Safety, RD
NCT03475004	II	Pembrolizumab + bevacizumab + binimetinib	Safety
NCT03396926	II	Pembrolizumab + bevacizumab + capecitabine	DLT, ORR
NCT04776148 (MK-7902-017/E7080-G000-325/LEAP-017)	III	Pembrolizumab + lenvatinib vs. SOC (Regorafenib or TAS-102)	OS
NCT05035381	II	Pembrolizumab + FOLFIRI + bevacizumab	ORR
NCT02298959	I	Safety, RD	OS
NCT04745130	II	Sintilimab + regorafenib + cetuximab	ORR
NCT03712943	I	Nivolumab + Regorafenib	MTD
NCT04963283	II	Nivolumab + cabozantinib	DCR
NCT04362839	I	Nivolumab + ipilimumab + regorafenib	RD
NCT03475953	I/II	Avelumab + regorafenib	RP2D, ORR
NCT02997228	III	mFOLFOX6 + bevacizumab vs. atezolizumab vs. mFOLFOX6 + bevacizumab + atezolizumab	PFS
NCT02873195	II	Capecitabine + bevacizumab + atezolizumab vs. PBO	PFS
NCT04659382 (SIRTCI)	II	Atezolizumab + XELOX + bevacizumab + SIRT	9 months-PFS
NCT02777710 (MEDIPLEX)	I	Durvalumab + pexidartinib	DLT, ORR
NCT03555149 (Morpheus-CRC)	I/II	Atezolizumab + bevacizumab or regorafenib combinations	ORR
NCT03170960	I/II	Atezolizumab + cabozantinib	MTD, ORR
NCT03539822	I/II	Durvalumab + cabozantinib	MTD, ORR
NCT05485909	II	Toripalimab + regorafenib + RFA	ORR
NCT04110093	I/II	Nivolumab or camrelizumab or sintilimab or toripalimab + regorafenib	ORR, PFS
NCT04866862	II	Camrelizumab + fruquitinib	ORR
NCT04695470	II	Sintilimab + fruquitinib	PFS
NCT04194359	III	Xelox + bevacizumab + sintilimab vs. PBO	PFS
NCT04764006	II	Sintilimab + surufatinib	ORR
NCT05438108	II	SBRT + Xelox + sintilimab + bevacizumab	ORR, AEs
NCT04271813 (APICAL-CR)	II	Sintilimab + anlotinib	ORR
NCT04745130	II	Sintilimab + regorafenib + cetuximab	ORR
NCT05524155	II	Sintilimab + regorafenib + HAIC	ORR, AEs
NCT05292417	II	Sintilimab + fruquitinib + GM-CSF	PFS
NCT04948034 (RIFLE)	II	SABR+ tislelizumab + fruquitinib	ORR
NCT05314101	II	Tislelizumab + bevacizumab + TAS-102	PFS
NCT04924179	II	Tislelizumab + fruquitinib + SBRT	PFS
NCT04777162	II	Tislelizumab + anlotinib	ORR
NCT05435313	II	Tislelizumab + fruquitinib + HAIC	ORR
NCT04577963	I/II	Tislelizumab + fruquitinib	RP2D, AEs, ORR
NCT04579757	I/II	Tislelizumab + surufatinib	DLT, ORR

AEs: adverse events; DLT: dose-limiting toxicity; HAIC: hepatic arterial infusion chemotherapy; MTD: maximum tolerated dose; ORR: overall response rate; OS: overall survival; PBO: placebo; PFS: progression-free survival; RD: recommended dose; RFA: radiofrequency ablation; RP2D: recommended phase II dose; SABR: stereotactic ablative radiotherapy; SIRT: selective internal radiation therapy; SOC: standard of care.

## Data Availability

Not applicable.

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
