# Peer review of "The Interplay between Anti-Angiogenics and Immunotherapy in Colorectal Cancer"

_life, 2022, doi:10.3390/life12101552_

Round 1
Reviewer 1 Report
Brigida Anna Maiorano et al. provide a review regarding the interplay between anti-angiogenics and immunotherapy in colorectal cancer.
Points to be considered:
1) The rationale of why the authors came up with this review.
2) What is the information that is not exactly available that motivated the authors to come up with this information. What are the current caveats and how do the authors highlight the current research in answering them? If not they need to address in future directions.
3)The authors could provide a little more consideration of genomic directed stratifications in clinical trial design and enrollments.
4) The underlying message here is that more precision and individualized approaches need to be tested in well designed clinical trials – a challenge, but I would be interested in their perspective of how this might be done
5) The authors need to come up with a table to show the role of endothelial cells as Checkpoint For Immunological Patrolling in CRC to highlight their exact role.
6) The authors need to highlight what new information the review is providing to enhance the research in progress.
7)As is now well known, tumors grow and evolve through a constant crosstalk with the surrounding microenvironment, and emerging evidence indicates that angiogenesis and immunosuppression frequently occur simultaneously in response to this crosstalk. Accordingly, strategies combining anti-angiogenic therapy and immunotherapy seem to have the potential to tip the balance of the tumor microenvironment and improve treatment response (please refer to PMID: 34067631 and expand)
Author Response
We would like to thank the reviewer for their valuable comments and suggestions to our paper. Herein, we report the answers to their comments:
1) "The rationale of why the authors came up with this review.": Thank you. We better specified in the manuscript the rationale for coming up with our paper, that is summarizing the pre-clinical and clinical evidence regarding the combination of anti-angiogenics and immune checkpoint inhibitors and how they
could be included in the clinical practice for CRC patients.
2) "What is the information that is not exactly available that motivated the authors to come up with this information. What are the current caveats and how do the authors highlight the current research in answering them? If not they need to address in future directions.": Thank you. We better specified in the
manuscript the possible future directions of this combination, and added a table resuming the main ongoing trials in this field in the Discussion section.
3)"The authors could provide a little more consideration of genomic directed stratifications in clinical trial design and enrollments." Thank you. We more accurately specified the genomic/molecular features of patients included in the different clinical trials we reviewed in the Results section.
4) "The underlying message here is that more precision and individualized approaches need to be tested in well designed clinical trials – a challenge, but I would be interested in their perspective of how this might be done": Thank you. We further discussed the limitations and future direction of this research field in this malignancy.
5) "The authors need to come up with a table to show the role of endothelial cells as Checkpoint For Immunological Patrolling in CRC to highlight their exact role.": Thank you. We added such as a table, alongside a brief discussion about ECs and their checkpoint role for immune cells and immune response in
the Discussion section, also updating references.
6) "The authors need to highlight what new information the review is providing to enhance the research in progress.": Thank you. We better specified the possible future directions of this combination, and added a table resuming the main ongoing trials in this field in the Discussion section.
7)"As is now well known, tumors grow and evolve through a constant crosstalk with the surrounding microenvironment, and emerging evidence indicates that angiogenesis and immunosuppression frequently occur simultaneously in response to this crosstalk. Accordingly, strategies combining anti-angiogenic therapy and immunotherapy seem to have the potential to tip the balance of the tumor microenvironment and improve treatment response (please refer to PMID: 34067631 and expand)": Thank you for the suggestion. We added the suggested reference and argued the topic in the Discussion section.
Reviewer 2 Report
The manuscript by Maiorano et. al is a timely and well organized review. The article discusses about the association of antiangiogenic therapy plus imune checkpoint inhibitors in MSS mCRC. Also, the reference list shows the expertize of the authors in the subject. I think that the review should be published in it's current form.
1. The research identifies trials that are focusing on targeted therapies ( anti0-angiogenic) plus immunotherapy.
2. The topic is relevant and of interest to medical oncologists that treat gastrointestinal tumors. It is of interest especially because the subject treats MSS mCRC patients for which therapeutic options like immunotherapy are lacking.
3. There aren't many published papers about the subject in the literature taking into consideration that the authors discussed about randomized clinical trials focusing on the subject.
4. Being a narrative review there isn't any concern about the methodology.
5. The conclusions are consistent and fully depict the main question.
Author Response
We are extremely grateful to the reviewer for their comments and opinion regarding our manuscript.
Round 2
Reviewer 1 Report
I am satisfied with the revisions provided and with the rebuttal.